# Analysis of Compliance with Proper Nutrition Principles in Patients with a History of Myocardial Infarction

**DOI:** 10.3390/nu16183091

**Published:** 2024-09-13

**Authors:** Patrycja Krężel, Ewa Kurek, Anna Jurczak, Izabela Napieracz-Trzosek, Dorota Iłgowska, Katarzyna Młyńska, Sylwia Wieder-Huszla

**Affiliations:** 1Department of Specialized Nursing, Pomeranian Medical University in Szczecin, 71-210 Szczecin, Poland; anna.jurczak@pum.edu.pl (A.J.); izabela.napieracz.trzosek@pum.edu.pl (I.N.-T.); sylwia.wieder.huszla@pum.edu.pl (S.W.-H.); 2Intensive Cardiac Care Unit, Independent Public Voivodship Hospital, ul. Arkońska 4, 71-455 Szczecin, Poland; goldberry.ewa@gmail.com; 3Deparment of Nursing, State University of Applied Sciences in Koszalin, 75-582 Koszalin, Poland; d.ilgowska@pwsz-koszalin.pl; 4Students’ Scientific Society of Department of Specialized Nursing, Pomeranian Medical University in Szczecin, 71-210 Szczecin, Poland; katarzyna553@wp.pl

**Keywords:** diet, nutrition, myocardial infarction, education

## Abstract

Adherence to dietary recommendations and the implementation of appropriate dietary habits after myocardial infarction (MI) can significantly improve health and reduce mortality from cardiac causes. The aim of this study was to analyse the adherence of patients with a history of MI to a healthy diet, which is one of the primary methods of cardiovascular disease (CVD) prevention. Following a proper diet involves limiting the consumption of saturated fats, salt, alcohol, and simple sugars. It is recommended to follow the Mediterranean diet, which is based on whole grains, fruits, vegetables, and fish. This study involved 120 patients hospitalised in the Invasive Cardiology and Cardiac Intensive Care Unit at the Independent Public Voivodship Hospital in Szczecin from August to December 2019. A self-designed questionnaire and the Questionnaire of Eating Behaviour (QEB) were used. The majority of respondents were hospitalised for a first-time MI (88.33%), and 65% of them reported adherence to the recommendations. The vast majority (78.33%) considered their diet good, with 50.83% of the respondents eating four meals a day and never eating fast food (49.17%). The analysis showed that although the respondents’ diets did not contain many unhealthy foods, they did not consume enough vegetables, fruits, fish, nuts, or fibre, which have a protective effect, lowering the risk of cardiovascular diseases and death. Furthermore, patients with a better education had a higher level of knowledge. Respondents’ knowledge of proper post-MI nutrition was sufficient, and their index of unhealthy diets was low, but they still made dietary mistakes and did not consume enough health-protective foods. These results indicate a need for further education.

## 1. Introduction

Myocardial infarction (MI) refers to the death of myocardial cells caused by ischaemia. It is important to distinguish between myocardial injury and MI. Both conditions are characterised by elevated cardiac troponin values; however, MI can be diagnosed when one of the following symptoms is present: a dynamic increase or decrease in troponins, symptoms of ischemia (usually chest pain and pressure radiating to the upper extremities, lower jaw, and upper abdomen, independent of exercise and body position; shortness of breath; fatigue; etc.), ECG changes indicative of ischaemia, or the identification of newly formed cardiac contractility abnormalities [1,2]. Although the prevalence and mortality of heart disease is declining, according to data from the World Health Organisation, 7.4 million people died from CHD in 2015, and it remains one of the most common causes of death worldwide [3,4].

People who have experienced myocardial infarction are at greater risk of death from cardiovascular causes. In one study, a higher mortality rate was observed in post-MI patients even if one year had elapsed since the onset of acute coronary syndrome. The 8-year mortality rate was as high as 59% [5]. To prevent the further development of CVD and reduce the risk of death, patients must try to reduce their exposure to risk factors. Not all risk factors can be influenced, as some factors are non-modifiable (age > 55 years in men and >65 years in women, male gender, or a family history of CVD). However, there are many modifiable factors that patients can control by adopting health-promoting behaviours to limit their negative effects [6,7]. According to the WHO, there are many important risk factors that depend on a person’s health behaviours. An unhealthy diet, along with a lack of physical activity, smoking, and alcohol consumption, is listed as one of the most important risk factors [8]. A proper diet is not only based on caloric intake; rather, the nutritional value of the foods consumed and the number of meals consumed in a day also play key roles. The diet that patients should follow in order to reduce the risk of developing and worsening CHD ought to include plenty of vegetables, fruit, whole grains, and plants, which have essential vitamins and minerals and also provide a high amount of fibre. This is because fibre has a cholesterol-lowering effect. Vegetable oils also have a similar effect and are therefore often recommended for patients at risk of MI. Meat should be consumed in moderate amounts, and it is advised to replace meat with fish. The consumption of red meat, on the other hand, is discouraged altogether [1,6,9].

In summary, dietary habits have a major impact on CHD, and changing them can significantly improve the health of post-MI patients. Therefore, this study aimed to assess the diets of cardiac patients and their adherence to dietary recommendations. Such evaluations could be helpful in identifying deficiencies in patients’ knowledge and could guide the planning of nutritional education initiatives based on the assessment results.

## 2. Materials and Methods

### 2.1. Study Design and Population

This study involved 120 patients (62 women and 58 men) admitted to the Ward of Invasive Cardiology and the Intensive Cardiac Care Unit at the Independent Public Voivodship Hospital in Szczecin. Data were collected from August to December 2019 using a diagnostic survey method with a self-designed questionnaire and a standardised tool. All respondents were informed about the aim of the study and that their data would only be used for research purposes. This study was completely anonymous, and participation was voluntary.

### 2.2. Applied Measurements

A self-designed questionnaire was used to collect sociodemographic and anthropometric data, as well as information on each patient’s cardiac conditions and other comorbidities. The standardised tool used in the study was the Questionnaire of Eating Behaviour (QEB). This tool is a Polish scale that examines eating habits, which specific foods are being consumed, the frequency of meals, opinions about nutrition, respondents’ own assessment of their diet, and respondents’ knowledge about nutrition and how they have cultivated it [10]. This questionnaire collects information on the frequency with which respondents consume products from 16 food groups. Respondents choose from six consumption frequency categories, ranging from “never” to “several times a day”, and their choices are then converted into daily frequency values (times/day). This results in two diet quality indexes, each of which includes eight food groups with either beneficial or non-beneficial effects on health. These indexes are the so-called healthy diet index, which includes vegetables, fruit, wholemeal and Graham bread, dairy products, fish preserves and fish-based dishes, and legumes, and the unhealthy diet index, which includes sweets, fried foods, alcoholic beverages, sweetened carbonated and energy drinks, canned meat, fish, and vegetables, instant soups and thickeners, and fast food [11].

### 2.3. Data Analysis

Statistical analysis was performed using the statistical package PQStat, specifically version 1.6.6.202. Food and nutrition knowledge in relation to education was analysed using the Kruskal–Wallis test and a post hoc Dunn’s test with Bonferroni correction. Additionally, the Jonckheere–Terpstra trend test was performed. The relationship between age and nutrition knowledge was analysed by estimating Spearman’s monotonic correlations. To assess food and nutrition knowledge in relation to gender, the Mann–Whitney U test was used. A test probability of *p* < 0.05 was considered significant, and *p* < 0.01 was considered highly significant.

## 3. Results

The study group had a mean age of 65.12 years, a mean height of 168.92 cm, and a mean weight of 81.27 kg. The average BMI of the respondents was 28.39. The gender distribution in the group was relatively even: 51.67% of the respondents were women, and 48.33% were men. Overall, 60% of the respondents were married. Half of the respondents reported that they had finished secondary education. In terms of work activity, 61.67% of the respondents were retired (Table 1). In the study group, cardiovascular diseases (CVDs) were the most common (58.33%), and for 88.33% of the respondents, the reason for hospitalisation was a first-time MI (Figure 1). Adherence to medical advice was reported by 65.83% of respondents. Cardiovascular diseases predominated in the study group, with 58.33% of respondents having reported them. The next most frequent conditions were osteoarticular (21.67%) and ocular diseases (21.67%). Only 14.17% of respondents stated that they had no comorbidities (Table 1).

The vast majority (88.33%) of respondents were hospitalised for a first-time MI. In more than half of these cases (55.83%), MI occurred in under six months, and in 67.5% of patients, percutaneous coronary intervention (PCI) placement was used as a form of treatment. Most patients (60.83%) underwent cardiac rehabilitation, and 65.83% declared that they followed their doctor’s recommendations regarding pharmacotherapy, regular visits to the outpatient clinic, and blood pressure monitoring. However, only 52.5% of the respondents took measurements of their blood pressure (Table 2).

### 3.1. Analysis of Eating Habits, the Products Consumed, and the Frequency of Consumption

Overall, 51% of patients report eating four meals a day, most often at irregular times (48%). As many as 80.83% of participants snack between meals (Table 3), and they snack frequently, ranging from eating between meals several times a week (36%) to multiple times a day (29%) (Figure 1). More than half of the respondents (53%) admitted that they sometimes add extra salt to their food. The average index of a healthy diet was 3.30 (SD 1.38), whereas that of an unhealthy diet was, on average, 1.65 (SD 1.07). The higher the index value, the higher the intensity of the feature. The intensity of both pro-healthy and unhealthy diets was low in the vast majority of cases (90% and 98%, respectively). This means that the diet of cardiac patients is only slightly unhealthy but sorely lacks foods with protective nutritional properties. In terms of how participants assembled their knowledge on nutrition, most of the patients indicated that they gained their understanding of nutrition at home (69.17%) (Table 3).

In addition to knowing what food products are being consumed, it is important to know how often respondents are eating. The diets of the respondents were quite diverse, as they rarely consumed selected foods every day. About half of the respondents consumed fruit, vegetables, and dairy products of different kinds several times a week. Sweets were also frequently consumed, with 40% of patients eating them multiple times a week. The respondents rarely drank alcohol. In total, 52% stated that they only drink one to three times a month. About half of the participants never drink carbonated drinks (51%) and never eat fast food (49%), and 88% never consume energy drinks. However, many respondents do not consume enough legumes and fish. These are most often eaten one to three times a month. Detailed data on the frequency of consumption of specific products are presented in Table 4.

It was also important to determine the specific types of products that were most frequently selected from the distinguished food groups. The type of products chosen, such as the most frequently consumed type of meat, determines whether this dietary component can be considered recommended or discouraged. A breakdown of the types of products chosen by respondents is shown in Figure 2.

### 3.2. Respondents’ Knowledge of Nutrition

Furthermore, an assessment of the patients’ knowledge about food and nutrition was conducted. The average score was 11.94 points out of a total of 25, which can be interpreted as a sufficient level of knowledge. The higher the index value, the greater the intensity of healthy or unhealthy eating habits. Values between 0 and 5.33 correspond to a low intensity of eating behaviours. Thus, both healthy and unhealthy diet intensities were low among the respondents. The intensity of the healthy diet index was low for 90% of the respondents, whereas up to 98% of respondents had a low intensity score for the unhealthy diet index (Table 5).

The correlation between education and knowledge of food and nutrition was the only statistically significant one (*p* < 0.05) (Table 6). The Jonckheere–Terpstra trend test indicated a significant increasing trend (Z = 2.2903, *p* = 0.0220) showing that as the level of education increases, so does the level of knowledge. The Kruskal–Wallis test (H = 9.2423, *p* = 0.0262) also showed a significant correlation; however, the post hoc test did not indicate significant differences between specific groups.

## 4. Discussion

This study attempted to analyse the opinions and dietary habits of post-MI patients and their knowledge of nutrition. Patients with coronary artery disease (78.33%) mostly assessed their diet as good and were shown to have a sufficient level of nutritional knowledge. Interestingly, more than half of the respondents (53%) also considered their level of knowledge as sufficient. This means that many patients accurately assess their own knowledge. However, our study showed that both the healthy diet index and the unhealthy diet index were low, which means that although patients consume relatively few unhealthy products, they do not include enough foods with cardio-protective effects. Other studies indicate that patients’ knowledge of nutrition is relatively low and that the same is true regarding adherence to dietary recommendations. In a study conducted by Alanazi A. et al., it was found that patients mainly associated smoking and obesity with the occurrence of MI; however, they did not recognise nutrition as an important factor [12]. The results of a study conducted in Pakistan were similar. Patients often have insufficient knowledge about the importance of a healthy diet, and although there was an improvement in the dietary habits of patients after MI, they still did not consume enough products with cardio-protective effects [13]. In addition, other studies have shown that, after MI, patients often do not change their diet, even after receiving education on the subject [14,15]. This correlates with the results obtained in this study, as 65.83% of patients reported continuous adherence to the recommendations, but this was not reflected in the survey results. However, by assessing patients’ preferences and eating habits and analysing their current knowledge and level of compliance, it is possible to design a personalised educational program for each patient.

It should also be noted that the mean BMI of the respondents was 28.39 (SD 4.72), a score indicating overweight. This may suggest that the respondents encounter difficulties implementing an adequate diet despite suffering from coronary heart disease, which is much more likely to affect overweight people [16]. Interestingly, recent studies report that patients with a higher BMI have a lower mortality rate, with such results also being reflected in the long-term prognosis of survival [17,18].

Half of the patients (50.83%) reported that they eat four meals a day. However, 47.5% of the respondents stated that they eat irregularly, and as many as 80% admitted to snacking. In terms of main meals, patients most often consumed bread, meat, soups, potatoes, and salads. The most common beverage of choice was tea. These are not foods that have the potential to reduce the risk of another MI. Furthermore, irregular eating promotes snacking between meals and weight gain. People who eat two to four meals at regular times show a reduction in cardiovascular disease symptoms and are more likely to be leaner. Meal irregularity, especially skipping breakfast and eating late in the evening, is associated with an increased risk of CVD and diabetes [19].

The majority of respondents add salt to their food, with 8% doing so frequently and 53% doing so sometimes. Studies show that high salt intake can lead to increased mortality in patients after MI. This is because high salt intake causes inflammation and structural changes in the heart muscle [6,20,21]. However, some studies suggest that using salt in adequate amounts reduces the risk of subsequent MI compared to a low-salt diet [22].

The respondents rarely eat fast food and drink alcohol, usually doing so one to three times a month or never. This reflects concern for one’s health, as avoiding these items has been repeatedly shown to reduce the risk of CVD and many other chronic diseases [6,23]. In addition, the respondents tend to consume low-alcohol beverages, such as beer or wine. Thus, they do not consume doses of alcohol greater than 100 g, which is considered safe. However, it should be noted that prior studies have shown that there is no safe dose of alcohol, that any amount of alcohol has a negative effect on blood pressure, and that alcohol consumption increases the risk of CVD [1]. However, other studies have shown that regular consumption of alcohol, in small amounts, could possibly have a positive effect on those with coronary heart disease, albeit only when combined with physical activity and a healthy diet [24].

This study found a significant correlation (<0.05) between education level and knowledge of food and nutrition. As patients’ level of education increased, so did their level of knowledge. A number of studies corroborate this correlation and also show that patients with a higher level of education learn faster [15,25]. Furthermore, many studies have found that older age, female gender, and higher socioeconomic status correlate with better knowledge and adherence to a healthier diet [15,26,27].

In summary, though the patients we studied know the basic principles of nutrition and avoid products commonly regarded as unhealthy, their diets do not contain ingredients that could reduce the risk of another CVD event, reinfarction, or death. This study demonstrates the continuous need for education in this area to improve patients’ eating habits and increase their chances of avoiding further dangerous CVD consequences. By assessing a specific patient’s knowledge and eating habits, educational programs including information relevant to the patient can be designed, which may increase their effectiveness. Improving a patient’s nutritional knowledge could also prompt them to exclude foods that may falsely be considered healthy from their diet and encourage them to introduce more cardio-protective foods into their diet.

## 5. Conclusions

The patients correctly assessed the state of their nutritional knowledge as sufficient, which was confirmed by the survey. This shows that they were aware of their own level of knowledge. However, the fact that their knowledge was only sufficient means that further education on the importance of following a healthy diet after MI should be provided. In this regard, the diet index could be a useful tool, as it allows one to precisely assess how healthy a person’s diet is. Our research showed that even though the patients’ diets did not include many unhealthy foods, they also did not include products with cardio-protective effects. Many of the respondents were overweight, which also suggests that they have problems adhering to an adequately nutritious diet. However, not many patients were actually obese. Therefore, in light of this recent research, whether significant weight loss should be advised should be considered. Furthermore, it was shown that the level of nutritional knowledge depends on the educational background of the participants. This means that the education provided should be personalised and that information should be communicated in a way that ensures it is easily understandable. This study highlights the need for education tailored to individual patients after a thorough assessment of their health, nutritional preferences, and the knowledge they already possess. The tool used in this study allows one to very accurately determine the types of food consumed, the frequency with which food is consumed, and patients’ knowledge about nutrition. This research also has some limitations, chief among them being the small size of the study group. Furthermore, the majority of patients were admitted to the hospital with a first-time MI, resulting in a less divided group. Additionally, some data, such as MI types, were not included in this study. However, the results of this study are valuable, and many researchers have also observed similar dependencies. However, due to the aforementioned limitations, the authors recognise the need for further research in this area.

## Figures and Tables

**Figure 1 nutrients-16-03091-f001:**
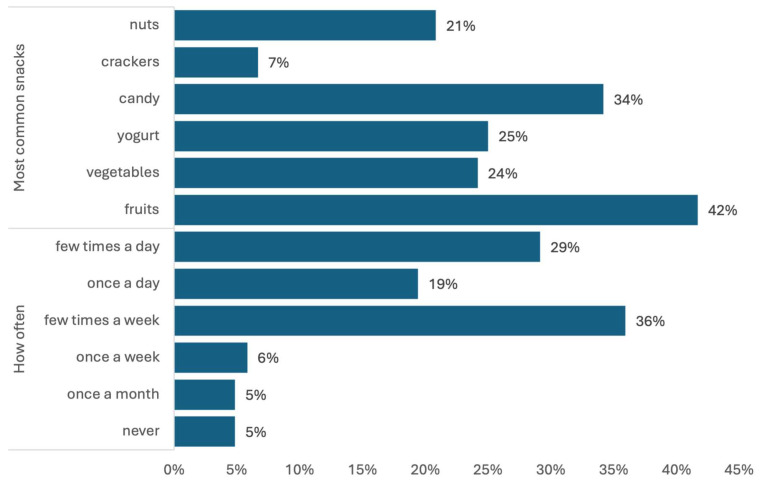
Foods most commonly eaten between meals and the frequency of snacking.

**Figure 2 nutrients-16-03091-f002:**
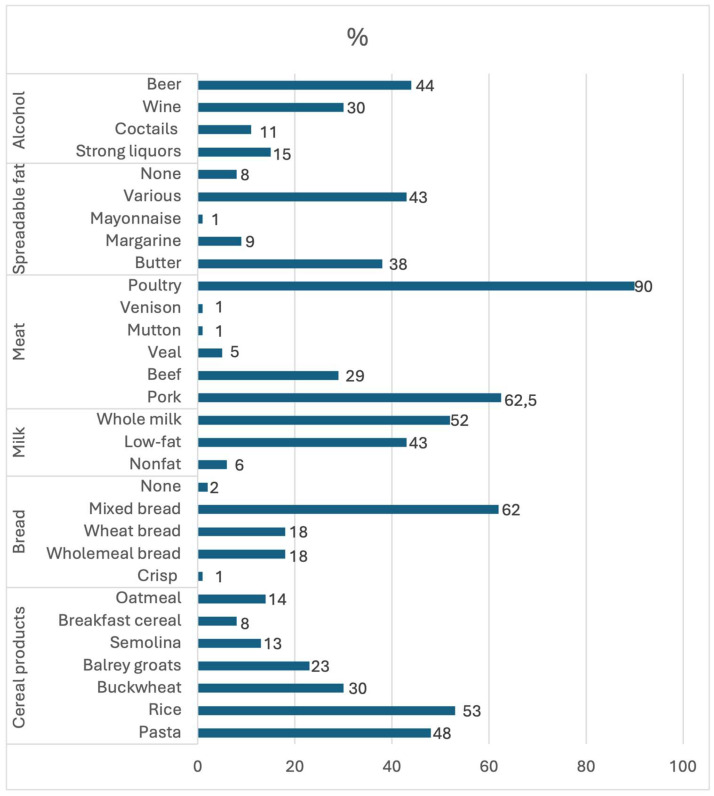
Types of selected products most often chosen by the respondents.

**Table 1 nutrients-16-03091-t001:** Sociodemographic and anthropometric data.

Variables		n (%)
Gender	Female	62 (51.67)
	Male	58 (48.33)
Marital status	Single/widowed/divorced	39 (32.5)
	Married/cohabiting	81 (67.5)
Education	Primary	11 (9.17)
	Vocational	26 (21.67)
	Secondary	60 (50)
	Higher	23 (19.17)
Employment status	Active	41 (34.17)
	Retirement	74 (61.67)
	Pension	3 (2.50)
	Other	2 (1.67)
Place of residence	City	86 (71.67)
	Countryside	34 (28.33)
Living with family	Yes	96 (80.0)
	No	24 (20.0)
Age, m (SD)		65, 12 (10.56)
Height, m (SD)		168, 92 (9.12)
Weight, m (SD)		81, 27 (15.98)
BMI, m (SD)		28, 39 (4.72)
Most common comorbidities		
Circulatory System		70 (58.33)
Respiratory system		10 (8.33)
Digestive system		18 (15.0)
Osteoarticular system		26 (21.67)
Genitourinary system		23 (19.17)
Neurological		5 (4.17)
Cancer		4 (3.33)
Ocular		26 (21.67)
Auditory		13 (10.83)
Metabolic		18 (15.0)
Other		5 (4.17)
None		17 (14.17)

n—number, m—mean, SD—standard deviation, and BMI—body mass index.

**Table 2 nutrients-16-03091-t002:** Patients’ cardiac details.

Patients’ Cardiac Details			n (%)
Number of MIs		One	106 (88.33)
		Two	11 (9.17)
		More than two	3 (2.50)
Time since first MI	Up to six months		67 (55.83)
	A year		27 (22.5)
	Over a year		14 (11.67)
	Several years		12 (10.0)
Cardiac rehabilitation	Yes		47 (39.17)
	No		73 (60.83)
Cardiac interventions performed within a year	Angioplasty		15 (12.5)
	Pharmacotherapy		22 (18.33)
	Pacemaker		2 (1.67)
	PCI		81 (67.5)
Compliance to recommendations	If feeling unwell		41 (34.17)
	Consistently		79 (65.83)
Self-measurement of blood pressure	Yes		63 (52.5)
	No		57 (47.5)

**Table 3 nutrients-16-03091-t003:** Dietary habits and patients’ nutritional self-assessment.

Variables		n = 120
Number of meals per day, %	1	1
	2	5
	3	33
	4	51
	5 or more	10
Eating at regular times, %	Yes, always	14
	Yes, some meals	38
	No	48
Snacking, %	Yes	81
	No	19
Adding salt to dishes, %	Yes	8
	Sometimes	53
	No	39
Diet index, (times/day) M(SD)	Health-promoting	3.30 (1.38)
	Unhealthy	1.65 (1.07)
Intensity of the health-promoting diet index, %	Moderate	10
	Low	90
Intensity of the unhealthy diet index, %	Moderate	2
	Low	98
Self-assessment of diet, %	Very good	3
	Good	78
	Bad	18
	Very bad	2
Self-assessment of nutritional knowledge, %	Very good	2
	Good	34
	Sufficient	53
	Insufficient	12
Nutrition knowledge sources, %	Home	69
	Advertisements	36
	Internet	28
	Press	26
	Radio	19
	Doctor	15
	School	3

**Table 4 nutrients-16-03091-t004:** Frequency of consumption of selected foods.

Foods	Frequency of Consumption, %					
	A Few Times a Day	Once a Day	Several Times a Week	Once a Week	One to Three Times a Month	Never
Vegetables	14	18	58	6	3	0
Fruits	17	23	52	6	3	0
Legumes	0	2	12	27	57	3
Wholemeal bread	3	13	17	19	28	21
Canned meat	1	2	17	13	52	16
Fish	0	1	13	23	58	5
Hard cheeses	2	5	51	18	18	8
Quark cheeses	1	4	45	30	18	3
Milk	5	9	43	11	17	16
Dairy drinks	0	4	49	25	19	3
Fried dishes	0	6	45	22	26	2
Fast food	0	0	5	6	40	49
Preserves	3	11	68	13	5	1
Sweets	3	17	40	18	20	2
Alcohol	0	2	8	11	52	28
Energy drinks	1	0	3	3	5	88
Carbonated drinks	0	2	13	12	23	51
Fruit juices	2	8	32	25	27	7

**Table 5 nutrients-16-03091-t005:** Analysis of the QEB score and diet quality index.

Average QEB score, m (Mdn)	11.94 (12)
Diet quality index	
Index of a healthy diet, m ± SD	3.30 ± 1.38
Index of an unhealthy diet, m ± SD	1.65 ± 1.07
Intensity of healthy diet index	
Moderate, %	10
Small, %	90
Intensity of unhealthy diet index	
Moderate, %	2
Small, %	98

**Table 6 nutrients-16-03091-t006:** Correlation of nutritional knowledge with the education of respondents.

Kruskal–Wallis Test	H	9.2423			
	*p*	0.0262			
		Primary	Vocational	Secondary	Higher
POST HOC (Dunn Bonferroni)	Primary		1.0000	0.2071	0.8485
	Vocational	1.0000		0.0628	0.7115
	Secondary	0.2071	0.0628		1.0000
	Higher	0.8485	0.7115	1.0000	

## Data Availability

All data are contained in this article.

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
