# Peer review of "Analysis of Compliance with Proper Nutrition Principles in Patients with a History of Myocardial Infarction"

_nutrients, 2024, doi:10.3390/nu16183091_

Round 1
Reviewer 1 Report
Comments and Suggestions for Authors
In my opinion introduction should be improved with more information about healthy diet in patients with cardiovascular disease
Myocardial infarction is not due to "closure of the coronary artery", occlusion is a more appropriate.
In lines 46-48 "And there are many modifiable risk factors that can significantly affect the course of the disease. According to the WHO, one of the most important determinants is appropriate nutrition." What do the authors mean with "appropriate nutrition."?
" MI occurred in under six months, and in 67.5% stent placement was used as a form of treatment." instead of "stent", "percutaneous coronary intervention" is a better expression.
In line "65.83% declared that they followed their doctor's recommendations", which were the recommendations?
I do not see the novelty of the information and its utility in the clinical practice.
My suggestion would be to analyze the dietary hobbits according to "Specific foods and food groups", as presented in 2021 ESC Guidelines on cardiovascular disease prevention in clinical practice:
Discussion should be improved according to the new data that would be included.
Comments on the Quality of English Languagemoderate English revision is recommended.
Author Response
Comments 1: In my opinion introduction should be improved with more information about healthy diet in patients with cardiovascular disease
Response: Thank you for this suggestion, we improved the introduction.
Comments 2: Myocardial infarction is not due to "closure of the coronary artery", occlusion is a more appropriate.
Response: Thank you for this suggestion, we changed it.
Comments 3: In lines 46-48 "And there are many modifiable risk factors that can significantly affect the course of the disease. According to the WHO, one of the most important determinants is appropriate nutrition." What do the authors mean with "appropriate nutrition."?
Response: Thank you for this suggestion, we changed it. Good nutrition is a major factor in maintaining health and the basis for cardiovascular disease (CVD) prevention, particularly for those at risk of CVD. According to the ESC Guidelines on cardiovascular disease prevention in clinical practice, a healthy diet refers to the nutrients and foods that are important for cardiovascular health.
Comments 4: " MI occurred in under six months, and in 67.5% stent placement was used as a form of treatment." instead of "stent", "percutaneous coronary intervention" is a better expression.
Response: Thank you for this suggestion, we changed it.
Comments 5: In line "65.83% declared that they followed their doctor's recommendations", which were the recommendations?
Response: Thank you for this suggestion, we changed it.
Comments 6: I do not see the novelty of the information and its utility in the clinical practice.
Response: Education of the society is an important element of preventive measures, and learning about food preferences and monitoring the diet of people from risk groups creates the possibility of a more conscious influence and therapeutic impact on the quality of their diet, by forming appropriate eating habits in these groups. Food can have a beneficial or negative impact on health, and thus reduce the risk of cardiovascular disease or reduce the risk of death.
Comments 7: My suggestion would be to analyze the dietary hobbits according to "Specific foods and food groups", as presented in 2021 ESC Guidelines on cardiovascular disease prevention in clinical practice:
Response: The authors supplemented the content in the indicated area.
Comments 8: Discussion should be improved according to the new data that would be included.
Response: The authors supplemented the content in the indicated area.
Quality of English Language - moderate English edition is recommended.
Response: The manuscript has been submitted for English editing.

Reviewer 2 Report
Comments and Suggestions for Authors
Abstract, line 18, Please provide a concise definition of a diet regarded as healthy.
Abstract, line 25, How could a marginally unhealthy diet also have a low protective factor?
Introduction, line 34, This sentence appears overly definitive. While occlusion of a coronary artery is a common cause of myocardial infarction, it is not the only cause. Please rephrase
Line 39, I would suggest supplementing the references with the current definition of myocardial infarction (DOI 10.1161/CIR.0000000000000617)
Lines 46-47, Please provide the appropriate references to support your statement.
Results, Authors should expand the baseline participants’ characteristics to include the type of MI, management (primary intervention etc), medication and comorbidities. These factors may directly influence participants' dietary habits. For instance, patients with STEMI might exhibit greater adherence to a healthy diet compared to NSTEMI patients, who are generally considered to have a less severe form of myocardial infarction. This difference could introduce selection bias and must be considered in the analysis.
Line 127, The term “healthy diet index” has not been previously introduced in the methods section. Hence, it is not clear what the authors imply. The same for pro-healthy and unhealthy diets.
Line 158, Please replace the term 'cardiac patients' with 'patients with coronary artery disease'.
Discussion, What are the implications of the study's findings? Please expand the discussion to include measures that could potentially enhance patients' adherence to healthier dietary habits.
Comments on the Quality of English LanguageVarious amendments are required.
Author Response
Comments 1: Abstract, line 18, Please provide a concise definition of a diet regarded as healthy.
Response: Thank you for this suggestion, we improved the abstract. According to ESC Guidelines on cardiovascular disease prevention in clinical practice a healthy diet plays a crucial role in preventing cardiovascular disease (CVD). A balanced diet (with lower intake of saturated fats, salt, carbohydrates, and alcoholic beverages) is recommended, with an indication of the Mediterranean diet or DASH due to their documented effectiveness in reducing the risk of CVD.
Comments 2: Abstract, line 25, How could a marginally unhealthy diet also have a low protective factor?
Response: Thank you for this suggestion, perhaps the sentence was confusing. We have changed the context of the statement. The consumption of particular groups of products, i.e. (vegetables, fruit, nuts, fish, fibre), promotes the regulation of cholesterol and blood pressure and thus reduces the risk of cardiovascular disease and decreases the risk of death. Therefore, we can speak of a protective effect.
Comments 3: Introduction, line 34, This sentence appears ovrly definitive. While occlusion of a coronary artery is a common cause of myocardial infarction, it is not the only cause. Please rephrase.
Response: Thank you for this suggestion, we changed it.
Comments 4: Line 39, I would suggest supplementing the references with the current definition of myocardial infarction (DOI 10.1161/CIR.0000000000000617).
Response: The authors supplemented the introduction in the indicated area.
Comments 5: Lines 46-47, Please provide the appropriate references to support your statement.
Response: The authors complemented the content of the introduction.
Comments 6: Results, Authors should expand the baseline participants’ characteristics to include the type of MI, management (primary intervention etc), medication and comorbidities. These factors may directly influence participants' dietary habits. For instance, patients with STEMI might exhibit greater adherence to a healthy diet compared to NSTEMI patients, who are generally considered to have a less severe form of myocardial infarction. This difference could introduce selection bias and must be considered in the analysis.
Response: Thank you for this suggestion, the text has been redrafted. The authors agree with the reviewer that the these factors may directly influence participants' dietary habits. Not all data were included in the conducted studies, therefore the authors see a need for further research in this area.
Comments 7: Line 127, The term “healthy diet index” has not been previously introduced in the methods section. Hence, it is not clear what the authors imply. The same for pro-healthy and unhealthy diets.
Response: Thank you for this suggestion, the authors have supplemented in the methods section the description of the tool with an explanation of the terms „pro-health diet” and “unhealthy diet”.
Comments 8: Line 158, Please replace the term 'cardiac patients' with 'patients with coronary artery disease'.Response: Thank you for this suggestion, we changed it. Comments 9: Discussion, What are the implications of the study's findings? Please expand the discussion to include measures that could potentially enhance patients' adherence to healthier dietary habits.
Response: The authors supplemented the discussion in the indicated area. Comments on the Quality of English Language - Various amendments are required. Response:
The manuscript has been submitted for English editing.

Round 2
Reviewer 1 Report
Comments and Suggestions for Authors
The authors made the suggested improvements.
All tables should respect the journal's recommendations.
Comments on the Quality of English LanguageEnglish is fine.
Reviewer 2 Report
Comments and Suggestions for Authors
The manuscript has been adequately revised